# Efficacy of the Piperidine Nitroxide 4-MethoxyTEMPO in Ameliorating Serum Amyloid A-Mediated Vascular Inflammation

**DOI:** 10.3390/ijms22094549

**Published:** 2021-04-27

**Authors:** Nathan J. Martin, Belal Chami, Abigail Vallejo, Albaraa A. Mojadadi, Paul K. Witting, Gulfam Ahmad

**Affiliations:** Faculty of Medicine and Health, School of Medical Sciences, The University of Sydney, Sydney 2006, Australia; nathan.martin@sydney.edu.au (N.J.M.); belal.chami@sydney.edu.au (B.C.); aval3972@uni.sydney.edu.au (A.V.); albaraa.mojadadi@sydney.edu.au (A.A.M.); paul.witting@sydney.edu.au (P.K.W.)

**Keywords:** nitroxide, 4-MethoxyTEMPO, serum amyloid A, endothelium, inflammation, atherosclerosis

## Abstract

Intracellular redox imbalance in endothelial cells (EC) can lead to endothelial dysfunction, which underpins cardiovascular diseases (CVD). The acute phase serum amyloid A (SAA) elicits inflammation through stimulating production of reactive oxygen species (ROS). The cyclic nitroxide 4-MethoxyTEMPO (4-MetT) is a superoxide dismutase mimetic that suppresses oxidant formation and inflammation. The aim of this study was to investigate whether 4-MetT inhibits SAA-mediated activation of cultured primary human aortic EC (HAEC). Co-incubating cells with 4-MetT inhibited SAA-mediated increases in adhesion molecules (VCAM-1, ICAM-1, E-selectin, and JAM-C). Pre-treatment of cells with 4-MetT mitigated SAA-mediated increases in transcriptionally activated NF-κB-p65 and P120 Catenin (a stabilizer of Cadherin expression). Mitochondrial respiration and ROS generation (mtROS) were adversely affected by SAA with decreased respiratory reserve capacity, elevated maximal respiration and proton leakage all characteristic of SAA-treated HAEC. This altered respiration manifested as a loss of mitochondrial membrane potential (confirmed by a decrease in TMRM fluorescence), and increased mtROS production as assessed with MitoSox Red. These SAA-linked impacts on mitochondria were mitigated by 4-MetT resulting in restoration of HAEC nitric oxide bioavailability as confirmed by assessing cyclic guanosine monophosphate (cGMP) levels. Thus, 4-MetT ameliorates SAA-mediated endothelial dysfunction through normalising EC redox homeostasis. Subject to further validation in in vivo settings; these outcomes suggest its potential as a therapeutic in the setting of cardiovascular pathologies where elevated SAA and endothelial dysfunction is linked to enhanced CVD.

## 1. Introduction

Acute phase response (APR) is a key component of the inflammatory response [1,2] and involves sequential physiological changes to combat inflammation, trauma or injury to host tissue [3]. During an APR, the body responds to maintain the internal equilibrium by metabolic and hormonal changes and among such adaptations, the regulation of plasma protein levels are of great significance. For example, C reactive protein (CRP) and serum amyloid A (SAA) are hepatically derived plasma proteins. In healthy individuals these proteins are normally present but at lower concentrations. However, during an APR the levels of CRP and SAA increase by up to 500- [4] or 1000-fold [3], respectively within 24–72 h. Currently, CRP is used as non-specific inflammatory marker for cardiovascular diseases [5], however, the biology and role of SAA is less known. SAA is produced in the liver in general and during an APR, de novo SAA is also synthesized by dysfunctional endothelial cells and other vascular cells [6,7].

Previously, we [8,9,10] and others [11,12,13,14,15] have shown a relationship between higher SAA levels and vascular pathology implicated in atherosclerosis. Although, accumulating evidence suggests that elevated serum SAA levels are associated with promoting pathology, the data is inconclusive. For example, De Beer et al. [16], have shown no link between the role of endogenous SAA and atherosclerotic lesion development in Apolipoprotein E-deficient (Apo-E^−/−^) mice. In humans, Yamada and Miida reported that SAA levels remain within physiological limits in patients with atherosclerosis [17]. On the other hand, in recent studies, SAA induced lung pathology is reported in Apo-E^−/−^ mice [18] and lower levels of SAA are reported to be associated with reduced risk of end stage renal disease (ESRD) in type 2 diabetic patients of American Indian ethnicity [19]. Moreover, others have reported that high levels of SAA show strong association with mortality rate in ESRD patients and support the argument that SAA is a better predictor of mortality in haemodialysis patients [20].

Notwithstanding this controversy described above, SAA is now documented to be detected in atherosclerotic lesions [14,21,22]. While the precise source(s) of the acute phase protein remains unclear, SAA is produced by endothelial/smooth muscles cells involved in atherogenesis [23] and SAA also modulates macrophages and monocytes function [24,25].

Physiological levels of SAA are essential for homeostasis but at pathological levels SAA leads to the production of reactive oxygen species (ROS) [26] and stimulates vascular cells to produce inflammatory cytokines [8] acting via cell surface receptors that activate intracellular signalling pathways implicated in stress processes. At abnormal levels, SAA induces oxidative stress by activating transcriptional factor NFκB [27,28], which leads to transcription of several inflammatory cytokines [28] involved in the APR. Marked increases in circulating SAA during APR in response to tissue injury/insult supports the notion that SAA may be a promising marker of cardiovascular risk [14] and mortality in ESRD patients [20], therefore, warranting further investigation on the biological action of this acute phase protein.

Taken together, targeting oxidative stress pathways via antioxidant and anti-inflammatory drugs has the potential to mitigate SAA mediated oxidative stress and hence downstream inflammatory response(s) leading to atherogenic process. Herein we report, therapeutic potential of cyclic nitroxides, 4-methoxy tempol (4-MetT), a class of novel cost-effective drug with dual properties, against SAA induced proatherogenic changes in cultured HAEC.

## 2. Results

### 2.1. 4-MetT Treatment Improves Cell Morphology

Morphological changes in cultured HAEC were detected in respective treatments 24 h post incubation with either DMSO (vehicle control), SAA (10 µg/mL) or SAA + 4-MetT (25 µM). SAA treated cells exhibited a marked loss in the initial cobblestone appearance and presented with an elongated and spindle shaped morphology compared to control (Figure 1A,B). Cultured HAEC pre-treated with 4-MetT resisted morphological change as cells appeared more similar to the control phenotype indicating that SAA induced structural changes were inhibited by 4-MetT treatment thereby, preserving cell integrity (Figure 1B,C). We also quantified the number of cells per treatment group that showed non-significant difference between the SAA and 4-MetT group. However, cell morphology changed and tended to become elongated in the presence of SAA compared to cells treated with 4-MetT or vehicle as control.

Considering these initial changes in cell morphology we further analysed the global expression of alpha-tubulin responsible for cell cytoskeleton in HAEC. Fluorescent imaging for alpha tubulin in SAA treated HAEC cells showed consistent disorientation in the pattern of tubulin expression with evidence of increased tubulin compared to control HAEC (Figure 2A,B). A higher proportion of multinucleated cells in SAA vs. control cells was also observed concomitant to the abnormal proliferation pattern identified in Figure 1. By contrast, cells treated with 4-MetT showed virtually no multinucleated cells however, the expression of alpha tubulin appeared to be greater than that detected in control cells (Figure 2C). Though difficult to interpret, we speculate that 4-MetT supports retention of cell structure through normalising alpha tubulin expression and distribution thus providing improved cytoskeletal integrity, cellular appearance and rate of cell proliferation in the presence of SAA-induced cellular stress.

### 2.2. 4-MetT Treatment Inhibits SAA Mediated Gene Expression of Inflammatory and Cell Adhesion Molecules

Gene expression of NF-κB, a common transcriptional factor of inflammatory mediators and adhesions molecules (e.g., VCAM-1, ICAM-1 and E-selectin) were studied by qPCR (Figure 3). Once an inflammatory response is established to added SAA (as evidenced by increased expression of NF-κB), modification of the junctional complexes and the proteins involved in junction formation are anticipated to be impacted and for this reason we investigated the levels of both NF-κB and downstream adhesion molecules.

Consistent with previous studies [8], NF-κB mRNA expression was 1.5-fold higher in cells exposed to SAA relative to the vehicle control after 6 h while co-administration with 4-MetT significantly inhibited SAA-stimulation of NFκB expression. These data indicate that 4-MetT can ameliorate the pro-inflammatory action of SAA (Figure 3A).

Similarly, the level of VCAM-1 (mRNA) expression in cells exposed to SAA alone measured 6 h after treatment was ~150-fold higher relative to the corresponding control (Figure 3B) indicating SAA-induced marked endothelial activation. In the presence of added 4-MetT (4-MetT+SAA), VCAM-1 mRNA expression showed a significant decrease compared to cells treated with SAA alone although this did not reach baseline (Figure 3B). Similar trends in the expression of E-selectin mRNA were observed in cells exposed to SAA in the absence or presence of 4-MetT (Figure 3C); ICAM-1 levels although decreased did not reach statistical significance (Figure 3D).

Considering the documented changes in the cell morphology and disoriented alpha tubulin expression (Figure 1 and Figure 2), we next quantified gene expression of selective cell-cell adhesion molecules. Consistent with the data for tubulin, mRNA expression of junctional adhesion molecule (JAM-C) was ~2-fold higher in SAA alone compared to the corresponding control, when assessed 6 h after treatment. By contrast, co-treatment with SAA and 4-MetT inhibited JAM-C expression compared to treatment with SAA alone (Figure 3E). Similarly, the gene expression of P120 Catenin increased ~1.5-fold compared to vehicle when cells were stimulated with SAA while P120 Catenin was significantly inhibited when cultured HAEC were co-treated with SAA and 4-MetT (Figure 3F).

### 2.3. 4-MetT Treatment Inhibits SAA Induced Protein Expression of VCAM-1, Caspase-1 and Improves cGMP Levels

Next, we investigated if expression of most relevant genes (VCAM-1) involved in atherosclerosis was translated into respective change(s) in protein. Western blot analysis of VCAM-1 in HAEC lysates showed ~80 (6 h) and ~60-fold (12 h) higher VCAM-1 protein expression in cells exposed to SAA alone relative to control (Figure 4). When cells were pre-treated with 4-MetT, SAA stimulated VCAM-1 protein expression was significantly reduced both at 6 and 12 h time points which corresponded closely to mRNA expression data indicating that 4-MetT treatment effectively inhibited VCAM-1 gene expression. Thus, 4-MetT minimised the impact of proatherogenic vascular cell adhesion in cells co-incubated with SAA and the nitroxide.

Regulation of NF-κB gene expression in SAA treated cells prompted us to investigate whether transcriptional activation of NF-κB (p65) was evident. Activated NF-κB is a regulator of Caspase 1 [29], through modulating Caspase-1 protein expression, which is then a mediator of downstream inflammatory responses and apoptosis [30]. The luminescence based inflammasome assay revealed ~4-fold increase in Caspase-1 levels in cells treated with SAA alone when compared to the control. Once again, co-incubation of cells with SAA + 4-MetT significantly inhibited the production of Caspsae-1 protein when compared to SAA alone, although the levels remained discernibly higher than control (Figure 5A).

The levels of cGMP, an indirect marker of vasodilation, were lower but non-significant in cells exposed to SAA alone relative to the control (Figure 5B). However, cells pre-treated with 4-MetT showed significant increase in the cGMP levels which directly aligns with the antioxidant capacity of cyclic nitroxide being a factor in preserving and even enhancing nitric oxide bioavailability/bioactivity.

### 2.4. Mitochondrial Function Studies

#### 2.4.1. 4-MetT Treatment Mitigates SAA Induced Mitochondrial Superoxide Radical Anion/ROS Production

Data collected to this point was consistent with SAA promoting a proinflammatory/heightened stress state that can lead to increased production of reactive oxygen species (ROS), which has been linked to apoptosis as supported by our assessment of Caspase-1. Accordingly, we measured mitochondrial superoxide radical anion/ROS levels in response to treatment with SAA in the absence and presence of 4-MetT using qualitative (fluorescent microscopy) and quantitative (absorbance) approaches with MitoSox Red probe specific to ROS and a Mito Tracker Green probe specific to mitochondria. Overall, the global levels of mitochondrial ROS production were higher in cells exposed to SAA alone compared to control as shown in fluorescent microscopy images (Figure 6A). Cells pre-treated with 4-MetT showed no or minimal MitoSox Red activation indicating that superoxide radical anion generation was effectively inhibited by 4-MetT. In parallel, quantitative analysis of MitoSox Red fluorescence showed similar patterns of decreased fluorescence where 4-MetT showed remarkable capacity to decrease mitochondrial ROS production to below that detected in the control cells (Figure 6B) completely aligning with real time fluorescent microscopy of cell treated under identical conditions (Figure 6A).

#### 2.4.2. 4-MetT Treatment Preserves Mitochondrial Membrane Potential

The mitochondrial membrane potential was determined via mitochondrial selective dye TMRM using both qualitative (fluorescent microscopy) and quantitative (absorbance) approaches in the presence of mitochondrial oxidative phosphorylation uncoupler (FCCP) as positive control. Cells treated with SAA alone showed reduced mitochondrial potential relative to control as early as 1 h of incubation which was improved in cells pre-treated with 4-MetT (Figure 7A) as indicated in fluorescent microscopic images. The quantitative analysis performed at 2 h revealed similar results showing significant reduction in mitochondrial membrane potential in SAA alone group which was rescued by pre-treatment with 4-MetT (Figure 7B).

#### 2.4.3. 4-MetT Treatment Restores Mitochondrial Respiratory Potential

The reduced mitochondrial membrane potential and demonstrable increase in mitochondrial ROS production prompted the analyses of mitochondrial respiratory potential using Seahorse XFe96 Analyser. Cells treated with SAA alone showed discernably higher rates in oxygen consumption than controls (Figure 8A) whereas, co-supplementation of HAEC with SAA and 4-MetT preserved oxygen consumption, although once again this was at a level higher than the corresponding control. In addition, cells treated with SAA alone tended to consume more energy and showed higher levels of maximal respiration relative to control and once again 4-MetT preserved levels of respiration and maximal respiration (Figure 8B,C). Concomitant to these results overall capacity to perform optimal respiration was lower in SAA alone group relative to control and showed a dramatic increase in the presence of 4-MetT (Figure 8D).

### 2.5. 4-MetT Treatment Improves Total Vellular NADP(H) Levels

Guided by mitochondrial respiratory studies, we next measured total cellular NADP(H) levels using luminescence based commercial NADP/NADPH- Glo™ Assay. Cells treated with SAA alone showed lower levels of total cellular NADP(H)/NADP(+) and NADP(+) relative to control but treatment with 4-MetT was not effective to bring these levels to control levels (Figure 9A,B). However total cellular NADP(H) levels were significantly improved by 4-MetT treatment as determined at 2 and 6 h post stimulation with SAA (Figure 9C). These data are in accord with the mitochondrial respiration experiments indicating that cells are experience elevated stress and tend to consume more oxygen that leads to higher generation of ROS.

## 3. Discussion

Serum Amyloid A (SAA) has the potential to be used as prognostic and diagnostic marker in a number of inflammatory disorders such as atherosclerosis, rheumatoid arthritis and end stage renal diseases [31]. Thus, significant increases in the circulating levels of SAA in response to APR is of direct clinical importance.

Outcomes generated here show for the first time that SAA treatment impacts mitochondrial function. In addition to providing energy to cell, mitochondria are also source of cellular ROS thus involved in oxidative stress as well as in programmed cell death [32]. Additionally, their non-energetic role in the regulation of innate immunity and inflammation is also reported [33]. We report 2.5-fold increases in the mitROS in response to SAA treatment that was significantly reduced with 4-MetT treatment. Increased mitROS can cause mtDNA damage [34] leading to mitochondrial dysfunction that has been linked in the development and progression of atherosclerotic lesions [35,36]. We also observed significant reduction in the mitochondrial membrane potential and significant increase in the mitochondrial respiration after SAA treatment which was again rescued by 4-MetT. These findings indicate that SAA impairs mitochondrial function and cell metabolism by reduced energy production, an event leading to apoptosis.

Nitroxides are a class of drugs with antioxidant and anti-inflammatory properties. Previously, we have shown that 4-MetT ameliorates dextran sodium salt (DSS) induced colitis in mouse model through its strong anti-inflammatory effects [37]. In the current study 4-MetT has significantly reversed the adverse effects of SAA on HAECs, highlighting its potential as a possible therapeutic agent acting through anti-inflammatory pathways. Interestingly, the 4-MetT treatment seems to restore the cell morphology that was disorientated after SAA treatment. For example, the number of multinucleated cells was reduced, and alpha tubulin integrity was maintained. Consistent with 4-MetT effectively decreasing SAA-stimulated proinflammatory state, drug-treated cells also exhibited decreased ROS production and improved mitochondrial function. While this mechanism of inhibiting ROS formation is active here, we cannot discount that 4-MetT is also acting through the nitroxide ability to act as a superoxide dismutase (SOD) mimetic [38] and dismutate superoxide radical anion [39] directly through this activity.

Tubulins are essential in maintaining the cell cytoskeleton that provide mechanical/architectural support to cell [40], cell division [41] and contribute to angiogenesis during plaque formation [41,42] and a compromises cell barrier will lead to increased vascular permeability leading to oedema [43]. Herein, we demonstrate that treatment with 4-MetT maintains tubulin integrity and cell morphology by reducing the SAA-induced stress response and at the same time supports cellular production of vasodilating NO thereby, ascribing potential anti-atherogenic activity to this nitroxide in addition to its anti-inflammatory actions.

SAA induced NF-κB expression shown here aligns with previous reports on human carotid artery endothelial cells [44], human atherosclerotic plaque [45] and murine intestinal epithelial cells [46]. While our results suggest SAA induced NF-κB activation in endothelial cells these data are limited as we measured the global expression of NF-κB and not the nuclear component which is true indicator of its transcriptional activity. Despite this limitation outcomes generated here did demonstrate the activation of several downstream pathways completely consistent with NF-κB transcriptional activation. NF-κB transcriptional activation in vascular endothelial cells, induces expression of an array of proinflammatory mediators (cytokines, chemokines) and cell adhesions molecules thus stimulating recruitment of monocytes that promote diseases development [47,48,49]. Our data aligns with previous reports that have shown nuclear translocation of NFκB subunit relA in the atherosclerotic lesions, smooth muscle cells and endothelial cells [50,51]. Therefore, targeting NF-κB -mediated inflammatory response could prove beneficial in mitigating the course of disease development and severity. For example, endothelial cell specific NF-κB inhibition in murine model of ApoE deficient mice have shown significant reduction in the atherosclerosis development [52].

Higher levels of VCAM-1/ICAM-1 are associated with atherosclerosis which take part in the disease progression as ICAM-1 help in transmigration of the immune cells in the sub-endothelial space [53]. VCAM-1 is major contributor in the initiation of atherosclerosis and is commonly detected in ~82% of vascular plaques [54]. In this study, the levels of VCAM-1 were significantly increased both at gene and protein levels in SAA treated cells. Outcomes from our data also demonstrate that 4-MetT consistently and significantly reduced the levels of NF-κB in treated cells compared to SAA alone. This activity resulted in a concomitant and significant reduction in the levels of VCAM-1 although this was not true for ICAM-1 (which decreased ~20% vs. SAA alone but remained non-significant). Whether 4-MetT has a direct inhibitory effect on these adhesion molecules or acts indirectly via inhibiting NF-κB or reducing ROS that is known to stimulate NF-κB remains unclear and warrants further investigation to establish the precise mechanism of action for this drug.

We also observed significant changes in the cell-cell adhesion molecules such as E selectin which is almost undetectable in healthy endothelium [55] but is expressed (2–8 h) in response to inflammatory stimuli [56,57]. E-selectin is strongly associated with the recruitment of leukocytes during initial phases of inflammation implicated in pathogenesis of atherosclerosis [56]. As with VCAM-1 and ICAM-1, expression of E-selectin is also mediated via the NF-κB inflammatory pathway [58] thus inhibition of NF-κB with 4-MetT employed here would indirectly inhibit E selectin expression, which diminishes SAA’s pro-atherosclerotic activity by limiting the rate of leukocytes adhesion to endothelial cells. An E selectin targeting approach in treating atherosclerosis has been reported by delivering mircoRNAs via microparticles [55]. Though 4-MetT showed significant reduction in the E selection levels in our study, the resulted are limited only to gene expression. Thus, a question whether 4-MetT inhibits expression of E selectin at protein level, more crucial for E selectin function, remains unanswered.

Data determined here also showed SAA-stimulated a significantly higher expression of JAM-C mRNA, a key player in the formation of endothelial tight junction complexes. JAM-C is involved in the trans-endothelial migration of neutrophils in vivo [59] and monocyte migration to endothelial cells in vitro [60]. Increased expression of JAM-C after treatment with SAA in our results means more cell polarisation promoting easy migration of the immune cell into the deeper tissue layers thus contributing to the lesions formation. Reduced transmigration of immune cells has been reported after JAM-C blockade in the human umbilical vein endothelial cells [61]. Therefore, 4-MetT treatment may be a potential approach in minimizing inflammatory cell recruitment via JAM-C blockade.

Another important factor responsible for the stability of endothelial tight junctions is the E-cadherin [62]. Stability of cadherin is dependent on binding of P120 with the E-cadherin to maintain cell barrier integrity [63,64]. Surprisingly, we observed significant increase in the P120 levels in SAA treated levels which were then reversed, even lower than normal levels, in 4-MetT treated cells. Though downregulation of P120 is reported in tumours when junctional control over cells is lost leading [65], the mechanism by which SAA has upregulated P120 levels here and how these raised levels are contributing to the inflammatory process are not known. Contradictory to our results, deletion of P120 has been linked with initiation of the inflammatory process [65] however, less is documented about the role of p120 in atherosclerosis. Therefore, SAA-mediated increase in the P120 levels in human endothelial cells, as we report here, requires further mechanistic studies to first validate these findings and subsequently understand the biological consequence(s) of this previously undocumented SAA activity.

Our results showed reduced levels of cGMP, an indirect marker of nitric oxide (NO) bioavailability, in SAA treated cells when compared to vehicle treated cells, but the reduction was not significant. Interestingly, 4-MetT treated cells showed significantly higher levels of cGMP compared to both SAA treated and vehicle treated cells. This indicates that 4-MetT is exerting its strong anti-inflammatory properties and by restoring the levels of NO it inhibits the caspase-1 activity thus lowering the inflammatory response. Apoptotic activity of Caspase-1 is related to nitric oxide (NO) concentrations in the cell [66]. Inhibition of NO shows increased caspase-1 mediated inflammatory response. On the other hand, inhibition of caspase-1 activity restores NO levels [67,68].

Taken together, we have shown that serum amyloid A shows strong pro-inflammatory properties. SAA stimulated cells show compromised cell morphology, increased production of NF-κB which then induces transcription of an array of pro-inflammatory cytokines and adhesion molecules. Further, SAA also triggers the apoptotic pathways by stimulating the caspase-1 activity indirectly reducing the levels of available circulating nitric oxide in conjunction with mitochondrial dysfunction. Treatment with 4-MetT showed significant inhibition or reversal of the SAA-induced negative effects on cultured cells. Though, we have shown promising results of cyclic nitroxides in combating the SAA induced inflammation, the effect of 4-MetT alone on the cells and exact mechanism of action of this drug was not studied in these experiments which warrants further mechanistic studies both in vitro as well as in vivo to explore whether the efficacy of 4-MetT can be translated in vivo.

## 4. Material and Methods

### 4.1. Materials

All materials were purchased from Sigma–Aldrich (North Ryde, NSW 1670, Australia) unless otherwise specified.

### 4.2. Methods

Human aortic endothelial cells (HAEC) (LONZA Inc. Walkersville, MD, USA) were cultured in supplemented EGM-2 media and maintained in a humified incubator at 370 C, 5% (*v*/*v*) CO2. All experiments were carried out on HAEC sub-cultured between passage 5–7. Data represent minimum of 3 replicates unless specified.

#### 4.2.1. Dose Selection

SAA (10 µg/mL) (recombinant human-Apo SAA) (PeproTech- Catalog #300-13) *SAA was made up in sterile filtered (0.22-micron filter) distilled water (d.H2O) and stored as aliquots at −30 °C. The dose chosen represents the value used in clinical settings (>8.9 µg/mL) that is able to activate experimental chronic inflammation [20]. Cyclic nitroxide 4-MetT (Sigma-Aldrich catalog #514586) was added at 25 µM in all experiments were used as this dose is phamacologically achievable [69]. Stock solution (100 mM) of 4-MetT was prepared in sterile-filtered (cell culture grade ‘Hybri-Max™’) dimethyl sulfoxide DMSO (Sigma-Aldrich catalog #D2650) and stored immediately at −30 °C.

#### 4.2.2. Treatment Duration

Cells were incubated with 4-MetT for 24 h prior to the addition of SAA in all instances. The cells were then incubated with SAA for various time points ranging from 2–24 h depending on the individual experiment carried out as specified in individual figure legends and methods. Briefly, addition of SAA/4-MetT was carried out in supplemented growth media in all instances. Media was aspirated from the cells only to be replaced with the specific requirement of the individual experiment to be carried, or for the collection of total cell protein lysates for protein and gene analysis (MPER™ lysis buffer (ThermoFisher Scientific, catalog #78501) Supplemented with cOmplete™, EDTA-free protease inhibitor cocktail (Roche, catalog #11873580001) and PhosSTOP phosphatase inhibitor (Roche, catalog #4906845001), 1X tablet each/50 mL lysis buffer.

#### 4.2.3. Treatment Groups

HAEC were classified into (i) control (DMSO), (ii) pre-treated with 4-MetT, 24 h prior to the addition of SAA (10 µg/mL) and (iii) SAA alone group (@10 µg/mL). In all experiments where cells were seeded at the specified density. Where required, cell viability and number were quantified using the Countess automated cell counter and using the “Trypan Blue exclusion” approach.

### 4.3. HAEC Imaging

At ~70% confluency cells were treated/incubated as explained earlier and imaged on a Zeiss AxioScope A1 (Carl Zeiss, Melbourne, Australia) under x 10 magnification and analysed using Zen Lite software (Carl Zeiss, Melbourne, Australia). Immunofluorescence imaging of HAEC was performed for alpha tubulin. Cells were cultured to ~70% confluence on Nunc™ Lab-Tek™ II Chamber Slides (Thermo Scientific, Melbourne, Australia). After respective treatments slides were washed 3 x with PBS, fixed in 100% ice-cold acetone for 30 min prior to the addition of rabbit anti-human primary antibody at a 1:50 *v*/*v* (Abcam, Sydney, Australia) prepared in antibody diluent (0.1% (*v*/*v*), Triton-X 100, 1% (*w*/*v*) BSA in TBS-T) and incubated overnight at 4 °C. Following the incubation period, the slides were washed 3 x TBST-T for 5 min and prior to the addition of the secondary IgG-Alexa Fluor^®^ 488 conjugated antibody (Life Technologies, Sydney, Australia) at 1:250 *v*/*v* dilution and incubated at room temperature for 45 min. The slides were again washed three times with TBS-T as above prior to the addition of SlowFade Gold Antifade Mountant with DAPI (Life Technologies, USA) and a glass cover slip. Finally, the slides were then imaged under fluorescent microscopy using a Zeiss Axio Scope A.1 (Carl Zeiss, Melbourne, Australia) under x 20 magnification and appropriate filter set. The captured images were then analysed using Zen2 Lite software (Carl Zeiss, Melbourne, Australia).

### 4.4. Gene Analysis (qPCR)

The primer sequence used for these studies is given in Appendix A. Total RNA was extracted from the cell lysates using the commercially available Isolate II Mini Kit (Bioline, Sydney, Australia) and cDNA was synthesized using sensiFAST reagent kit (Bioline, Sydney, Australia, catalogue #BIO-65053) with an Eppendorf MasterCycler gradient System (Eppendorf, Sydney, Australia). qPCR was carried out with 3.5 µL cDNA (synthesized using 300 ng total RNA/sample). Final primer concentration was optimized at 658 nm/reaction. A PCR reaction mixture was made for each gene analysed consisting of; PowerUp SYBR Green master mix (ThermoFisher Scientific, Waltham, MA, USA), d.H20 and the specific and forward reserve primer for the gene of interest. 6.5 µL of this rection mixture was added to each well of a 384-well qPCR reaction plate and finally 3.5 µL of cDNA (equating to 300 ng total RNA/sample) was added. The plate was sealed, placed into a centrifuge and spun at 300 r.p.m. for 5 min to spin down components. The plate was loaded into the LightCycler 480 II (Roche, Bern, Switzerland) and run according to the following thermal cycling conditions. 50 °C for 2 min for UDG activation, 95 °C for 2 min to activate DNA polymerase and followed by 45 repeating cycles of 95 °C for 15 s to denature, and 60 °C for 1 min to extend and anneal the sequence. A dissociation step was then run at the conclusion of every reaction to generate the melt curve as follows. 95 °C for 15 s at ramp rate of 1.6 °C/s, 60 °C for 1 min at ramp rate of 1.6 °C/s, and 95 °C for 15 s at a ramp rate of 0.15 °C/s. Amplification plots were then analysed using the LightCycler 480 software, to determine the threshold cycles (CT) and to check for nonspecific amplification via analysis of the melt curve. Gene expression for each sample was then quantified using the delta delta Ct method against the housekeeping gene, GAPDH.

### 4.5. Western Blot

Protein expression of vascular adhesion molecule-1 (VCAM-1; ab134047 Abcam), the most common adhesion molecule, in whole cell lysates was quantified by Western Blot. Following incubation period cells were harvested in M-PER™ lysis buffer as per the manufacture’s guidelines (ThermoFisher Scientific, Waltham, MA, USA) (supplemented with 1 x cOmplete™ EDTA-free protease inhibitor cocktail, 1 x phosSTOP phosphatase inhibitor (Roche, Bern, Switzerland). Total protein content of the cellular lysates was then determined using the bicinchoninic acid assay (BCA) with a bovine serum albumin (BSA) standard. The lysates were subsequently aliquoted and immediately stored at −80 °C for further analysis. A total of 15 µg protein/sample was separated by SDS-PAGE in a 12% *w*/*v* acrylamide gel (Bio-Rad) at 200 v for 1 h. Following separation, the gel was activated via UV for 5 min using the GelDOC XR+ (Bio-Rad). The gels were subsequently transferred onto a Trans-Blot^®^ Turbo™ Midi polyvinylidene difluoride membrane (PVDF) pre-assembled transfer pack (Bio-Rad) using the Trans-Blot^®^ Turbo™ Transfer System in a semi-dry rapid transfer for 15 min. The membranes were then blocked in a 5% skim milk (1 x TBS containing 0.01% *v*/*v* Tween-20 (TBS-T)) solution at room temperature for 1 h with gentle agitation. Following blocking stage, membranes were washed twice in TBS-T for 5 min/wash. The membranes were then incubated overnight at 4 °C with anti-VCAM-1 primary antibody (1: 2000) in 5% BSA/1x TBS-T. Membranes were washed three times in 1 x TBS-T for 5 min and incubated with an anti-rabbit IgG horseradish peroxide-conjugated secondary antibody (1: 5000; IgG-HRP: A6154 Sigma-Aldrich) in a 1% *w*/*v* BSA/1x TBS-T solution at room temperature for 1 h. Membranes were washed three time in 1 x TBS-T for 5 min/wash, and imaged using the Clarity ECL system (Bio-Rad) and ChemiDOC MP Imaging System as per the manufactures guidelines.

#### Re-Probing of Membranes for B-Actin Normalisation

Following imaging, the membranes were then stripped using a mild stripping buffer (0.015% *w*/*w* glycine, 0.001% *w*/*w* SDS, 0.010% *v*/*w* Tween 20, pH2.2) to remove ECL and previously bound antibody while preserving protein content. After a 10 min incubation with the stripping buffer, the buffer was replaced with fresh stripping buffer and incubated for a further 10 min. the membrane was washed twice in PBS for 10 min, and twice in TBS-T for 5 min. The stripped membranes were blocked in a 5% skim milk *w*/*v* (1 x TBS containing 0.01% *v*/*v* Tween-20 (TBS-T)) solution at room temperature for 1 h with gentle agitation. Following blocking stage, membranes were washed twice in TBS-T for 5 min/wash. The membranes were then incubated for 2 h at room temperature with anti-β-actin primary antibody (final dilution 1:2000 *v*/*v*; Cell-Signaling Technologies Australia) in 2% *w*/*v* BSA/1x TBS-T. Membranes were washed three times in 1 x TBS-T for 5 min and incubated with an anti-rabbit IgG horseradish peroxide-conjugated secondary antibody (final dilution 1: 5000 *v*/*v*) in a 1% *w*/*v* BSA/1x TBS-T solution at room temperature for 1 h. Membranes were washed three time in 1 x TBS-T for 5 min/wash, and imaged as outlined above. Data was analysed using Bio-Rad ImageLab™ (v6.0.1, 2017) with VCAM-1 expression expressed relative to β-actin and normalised against the control.

### 4.6. Caspase-GLO1 Assay

Caspase-1 activity levels were measured in cultured HAEC by a commercially available luminescent assay (Promega Corporation, Madison, WI, USA). Briefly, cells were seeded into a Corning™ 96-well White Microplate at previously optimised 40,000 cells/well and treated as above. The assay was run as per the manufacture’s guidelines with luminescence read every 15 min over a 60-min period using an Infinite^®^ M200 PRO Plate reader (Tecan, Männedorf, Germany). Data was recorded and analysed in Microsoft^®^ Excel^®^ (2013, v7) and normalised against total cellular protein following BCA protein assay on the same plate.

### 4.7. Cyclic Guanosine Monophosphate (cGMP) Assay

Intracellular levels of the secondary messenger signaling molecule cGMP were quantified via way of a commercially available ELISA kit (Abcam, Cambridge, UK). Whole cell protein lysates were thawed and centrifuged (10,000× *g*, 10 min, 40 C). The lysates were then used in the assay which was run as per the manufacture’s standard guidelines with absorbance recorded using an Infinite^®^ M200 PRO Plate reader (Tecan, Männedorf, Germany). All data was recorded and analysed in Microsoft^®^ Excel^®^ (2013, v7) and normalised against the previously determined total protein content.

### 4.8. Mitochondrial Superoxide (MitoSOX) Analysis

Levels of mitochondrial superoxide species (mtROS) were determined by use of the mitochondrial specific fluorescent probe MitoSOX Red (ThermoFisher Scientific, Waltham, MA, USA) in two independent assays; (1) fluorescent microplate reader, and data corroborated with (2) fluorescent microscopy.

#### 4.8.1. Fluorescent Microplate Method

In brief, cells were seeded at 30,000 cells/well onto a sterile 96-well black microplate (Corning, Costar) 48 h prior to experimentation. After 24 h media was refreshed, and the cells were treated as previously outlined with an additional control group of cells treated with 1 mM rotenone (Sigma Aldrich) for 2 h at 37 °C and 5% (*v*/*v*) CO2(g). Following the incubation period, MitoSOX™ Red reagent was reconstituted in sterile DMSO immediately prior to its use to yield a 5 mM stock solution which was further diluted in sterile HPPS at a final concentration of 5 µM. Subsequent to washing the cells in HBBS, this working solution of MitoSOX™ Red was added onto the cells and the plate incubated for 30 min at 37 °C and 5% (*v*/*v*) CO_2_(g). The cells were washed 3 times in HBSS, and the fluorescence was recorded at 510/580 nm, immediately and every 15 min over a 60-min period using an Infinite^®^ M200 PRO Plate reader (Tecan, Männedorf, Germany). Data was recorded and analysed in Microsoft^®^ Excel^®^ (2013, v7) and normalised against total cellular protein following BCA protein assay on the same plate.

#### 4.8.2. Fluorescent Microscopy Method

The experimental design generally followed what was outlined in (1) with cells seeded at 10,000 cells/well and no rotenone treatment was included in the assay. Following optimistaion of dose, MitoSOX™ was added at a final concentration of 1 µM to the cells with incubation and washing cycles as also described in (1). To confirm the MitoSOX™ fluorescent signal was selective for mtROS, the mitochondrial specific fluorogenic dye MitoTracker™ Green FM (ThermoFisher Scientific) was used to demonstrate co-localisaton of signal. MitoTracker™ was added to the cells at final concentration of 150 nM and the plate incubated for 10 min at 37 °C and 5% (*v*/*v*) CO_2_(g). The cells were then imaged under fluorescent microscopy using a Zeiss Axio Scope A.1 (Carl Zeiss, Melbourne, Australia) under x 20 magnification and appropriate filter sets. The captured images were then analysed using Zen2 Lite software (Carl Zeiss, Melbourne, Australia).

### 4.9. Mitochondrial Membrane Potentials Analysis

Cellular mitochondrial membrane potential (Δψm) was determined via measurement of the mitochondrial selective dye tetramethylrhodamine methyl ester (TMRM), Image-iT™ TMRM Regent (ThermoFisher Scientific, Waltham, MA, USA) in two independent assays; (1) Microplate read of absorbance at 548 nm, and, (2) via fluorescent microscopy.

#### 4.9.1. Microplate Absorbance Read at 548 nm

In brief, cells were seeded at 30,000 cells/well onto a sterile 96-well black microplate (Corning, Costar) 48 h prior to experimentation. After 24 h media was refreshed, and the cells were treated as previously outlined. An additional control group of cells were treated with the mitochondrial uncoupling agent carbonyl cyanide-p-trifluoromethoxyphenylhydrazone (FCCP) at final concentration of 1 µM for 10 min at 370 C and 5% (*v*/*v*) CO2(g). Following the incubation period, TMRM was reconstituted in sterile DMSO immediately prior to its use to yield a 100 µM stock solution which was further diluted in sterile PBS (0.02% *w*/*v* BSA) at a final concentration of 500 nM. Subsequent to washing the cells in PBS (0.02% *w*/*v* BSA), this working solution of TMRM was added onto the cells and the plate incubated for 30 min at 370 C and 5% (*v*/*v*) CO2(g). The cells were washed 3X in PBS (0.02% *w*/*v* BSA), and the absorbance was recorded at 548/574 nm, immediately and every 15 min over a 60-min period using an Infinite^®^ M200 PRO Plate reader (Tecan, Männedorf, Germany). Data was recorded and analysed in Microsoft^®^ Excel^®^ (2013, v7) and normalised against total cellular protein following BCA protein assay on the same plate.

#### 4.9.2. Fluorescent Microscopy

The experimental design generally followed what was outlined in (1) with cells seeded at 10,000 cells/well. Following optimistaion of dose, TMTM was added at a final concentration of 100 nM in HPPS (0.02% BSA) to the cells with incubation and washing cycles as also described in (1) supplementing HBPS/HPPS (0.02% BSA) for PBS. The cells were then imaged under fluorescent microscopy using a Zeiss Axio Scope A.1 (Carl Zeiss, Melbourne, Australia) under x 20 magnification and appropriate filter sets. The captured images were then analysed using Zen2 Lite software (Carl Zeiss, Melbourne, Australia).

### 4.10. Monitoring Mitochondrial Function

Cells were seeded at a pre-optimised 50,000 cells/well into an XF96 Cell Culture Microplate (Seahorse Bioscience) 48 h prior to experimentation. After 24 h media was refreshed, and the cells were treated as previously outlined. Prior to running the assay (at least 1 h) media was replaced with sterile supplemented Seahorse XF base media (containing: 10 mM glucose, 2 mM glutamine, 1 mM sodium pyruvate, pH 7.4) and incubated at 37 °C and 0% (*v*/*v*) CO2 (g). During this incubation period the XF96 Analyser was calibrated as per the manufacture’s guidelines with a pre-hydrated XF96 Sensor Cartridge plate containing calibration media. Following the calibration and incubation periods the cell culture microplate was loaded into the XF96 Analyser and a pre-designed assay template was run using Agilent Seahorse WAVE software (v2.6.1). OCR was measured at regular intervals following the automated injection of 1 µM oligomycin (Sigma), 1 µM FCCP (Sigma), and 0.5 µM antimycin A (Sigma) plus 0.5 µM rotenone (Sigma). Data was recorded and analysed in a combination of WAVE software (v2.6.1) and Microsoft^®^ Excel^®^ (2013, v7). All data sets were normalised against total cellular protein following BCA protein assay on the same plate.

### 4.11. NADP/NADPH Assay

Total NADP/NADPH, and individual NADP+ and NADPH levels were quantified via commercially available luminescent assay (Promega Corporation, Madison, WI, USA). Briefly, cells were seeded into a Corning™ 96-well White Microplate at a previously optimised 15,000 cells/well and treated as above. The assay was run as per the manufacture’s guidelines following the modified protocol for separate measurement of cellular NADP+ and NADPH with luminescence read at 30 and 60 min using an Infinite^®^ M200 PRO Plate reader (Tecan, Männedorf, Germany). Data was recorded and analysed in Microsoft^®^ Excel^®^ (2013, v7) and normalised against total cellular protein following BCA protein assay on the same plate.

### 4.12. Statistical Analysis

Data was presented as mean ± SD and analysed using GraphPad Prism software version 8 by ANOVA using Neumann–Keuls or Tukey comparison tests. A *p* value <0.05 was considered significant.

## Figures and Tables

**Figure 1 ijms-22-04549-f001:**
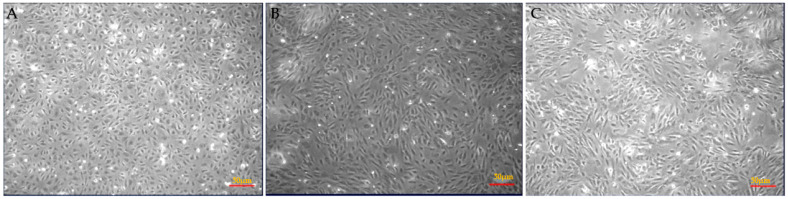
SAA-stimulates morphological changes in cultured HAEC. Representative images are shown for confluent HAEC treated with (**A**) DMSO (vehicle control) (**B**) 10 µg/mL SAA for 24 h, or (**C**) pre-treated with 25 µM 4-MetT for 24 h prior to addition of 10 µg/mL SAA and incubated further for 24 h. Data shown are representative of a single field of view, (**A**–**C**). Images were captured using an inverted light microscope with 10× magnification. Insets to panels (**A**–**C**) show higher magnification images of cells across the corresponding treatment groups.

**Figure 2 ijms-22-04549-f002:**
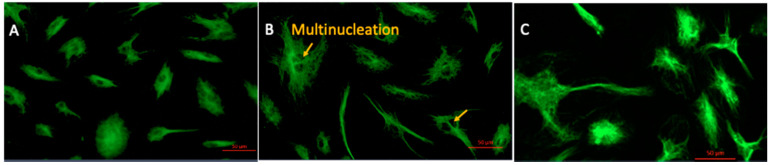
Changes in the cytoskeletal protein, alpha-tubulin in response to incubation of HAEC with SAA, with and without added 4-MetT. Representative images correspond to (**A**) Control (**B**) 10 µg/mL SAA (**C**)10 µg/mL SAA + 25 µM 4-MetT pre-incubation. Data shown are representative of (*n* = 3 control, *n* = 5 SAA, SAA + 4-MetT) independent studies.

**Figure 3 ijms-22-04549-f003:**
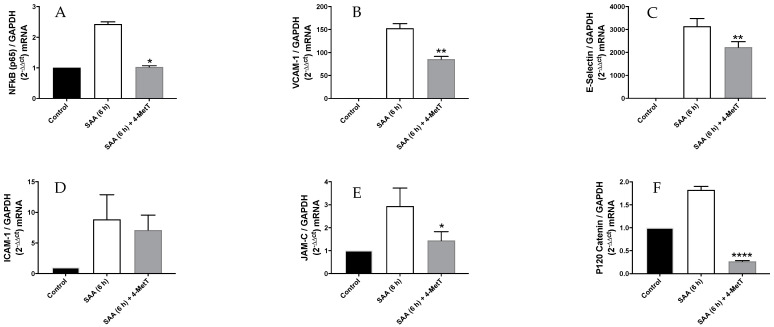
The effect of SAA and 4-MetT on pro-inflammatory, and adhesion molecule mRNA levels in cultured primary HAEC. Gene expression levels are shown for (**A**). NF-κB, (**B**). VCAM-1, (**C**). ICAM-1, (**D**). E-Selectin. (**E**). JAM-C. (**F**). P120 Catenin. Gene expression across treatment groups was normalised against GAPDH and expressed relative to the control using the semi-quantitative 2^−∆∆ct^method of analysis. Data represent mean ± SD, *n* = 3 in independent experimnets except VCAM-1, ICAM-1(n-2). * *p* < 0.05; ** *p* < 0.001; **** *p* < 0.0001 Different to SAA-treatment group.

**Figure 4 ijms-22-04549-f004:**
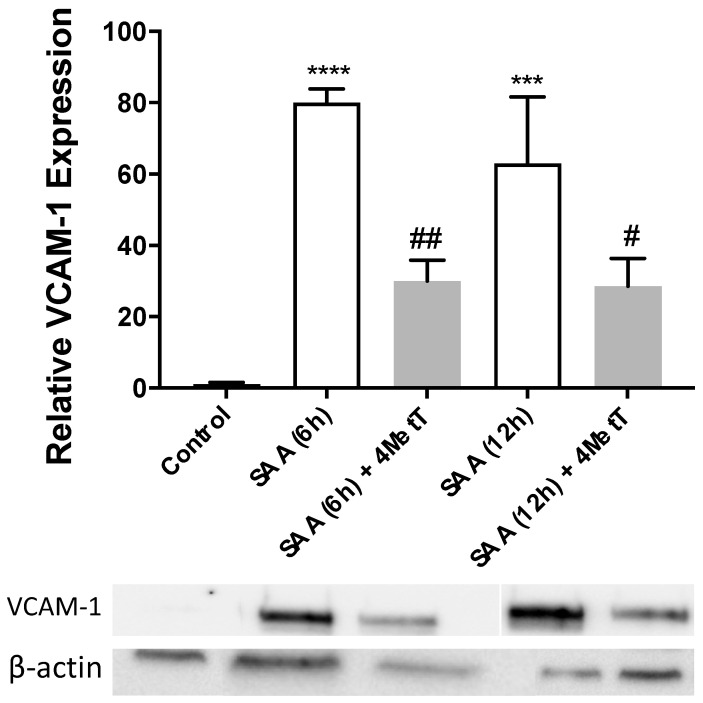
Effect of SAA and 4-MetT on vascular cell adhesion molecule-1 (VCAM-1) protein expression in cultured primary HAEC. VCAM-1 protein expression in HAEC was assessed by means of Western blot analysis. Lysates were generated from cells that were treated with 10 µg/mL SAA at either 6 or12 and with or without 25 µM 4-MetT preincubation for 24 h prior to adding SAA, or DMSO (vehicle control) and proteins were separated via SDS-page. The resultant membrane was then immunoblotted with an anti-VCAM-1 primary and secondary antibody pair. Data represent mean ± SD, *n* = 3 independent biological replicates. *** *p* = 0.0001; **** *p* < 0.0001. Different to control in the absence of SAA, ^#^
*p* < 0.05; ^##^
*p* < 0.001 Different to SAA-treatment group.

**Figure 5 ijms-22-04549-f005:**
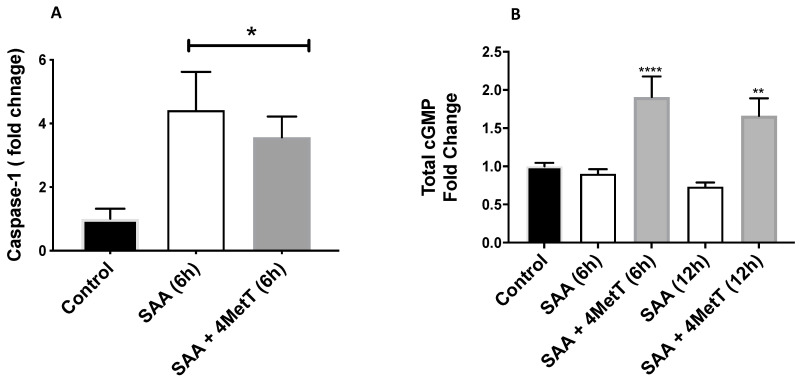
Caspase-1 levels in HAEC in response to SAA and 4-MetT. (**A**). Cells were treated with 10 µg/mL SAA at either 6 h and with or without 25 µM 4-MetT pretreatment for 24 h, or DMSO (vehicle control). Data represent mean ± SD, *n* = 7 control, SAA; *n* = 3 SAA + 4-MetT, independent biological replicates. * Different to control in the absence of SAA, *p* < 0.005 (**B**). Total cGMP levels in HAEC total cell protein lysates in response to added SAA and 4-MetT. * Different to control in the absence of SAA, * *p* < 0.05; ** *p* < 0.001; **** *p* < 0.0001Different to SAA-treatment group.

**Figure 6 ijms-22-04549-f006:**
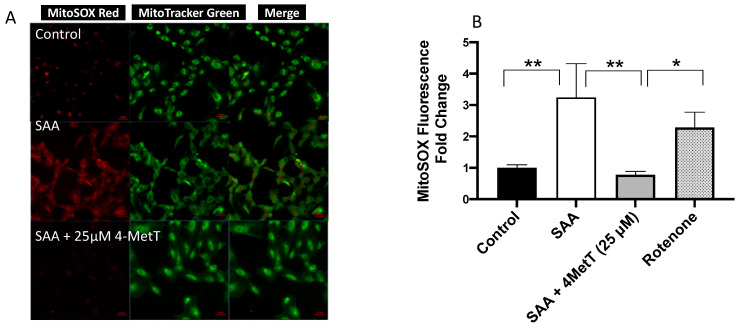
Mitochondrial ROS (mtROS) in cultured primary HAEC in response to SAA and 4-MetT. The mitochondrial specific fluorescent probe MitoSOX Red was used to quantify the production of mtROS in response to SAA and 4-MetT in HAEC (see Methods Section). Cells were treated with 10 µg/mL SAA at either 6 h and with or without 25 µM 4-MetT pretreatment for 24 h, or DMSO (vehicle control) in both independent runs. Rotenone was used as positive control that has known impact on cell respiration, and it confirmed that the experimental model was functioning as anticipated in the presence of the mitochondrial complex 1 inhibitor. (**A**). Images captured via fluorescent microscopy at ×20 magnification using an appropriate filter set. MitoTracker green was used as a control to fluorescently label mitochondria within the cell to show a co-localisation of the MitoSOX Red fluorescence (merge image; scale bar 20 µm; *n* = 4). (**B**). Fluorescence was measured at λ_excitation_/510/λ_emission_ 580 nm respectively after 30 min MitoSOX incubation using an Infinite^®^ M200 PRO Plate reader (Tecan, Männedorf, Germany). Data represent mean ± SD, *n* = 3 independent biological replicates, and were normalised against corresponding total cellular protein and expressed as relative fold change compared to the control (arbitrability assigned a value of 1). * *p* < 0.05; ** *p* < 0.005.

**Figure 7 ijms-22-04549-f007:**
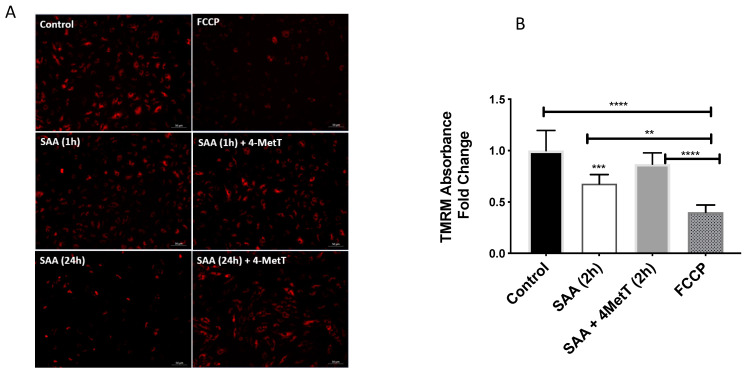
Effect of SAA and 4-MetT on mitochondrial membrane potential (Δψm) in cultured primary HAEC. TMRM was used to provide a measure of Δψm (see Methods Section). FCCP was used as a control to depolarise the mitochondrial membrane. (**A**). Cells were treated with 10 µg/mL SAA for either 2 or 24 h with or without 4-MetT pre-treatment for 24 h. Images captured via fluorescent microscopy at ×10 magnification using an appropriate filter set. Images are representative of X fields of view from *n* = 3 Control, FCCP; *n* = 4 SAA (2 h); *n* = 5 SAA (2 h) + 4-MetT, SAA (24 h), SAA (24 h) + 4-MetT independent experiments (scale bar 50 µm). (**B**). In a repeat independent study signal intensity was measured at 548/574 nm absorbance/emission respectively after 2 h incubation with respective treatments using an Infinite^®^ M200 PRO Plate reader (Tecan, Männedorf, Germany). Data represent mean ± SD, *n* = 6 independent biological replicates. All data was normalised against corresponding total cellular protein and expressed as relative fold change compared to the control (arbitrability assigned a value of 1). ** *p* = 0.003; *** *p* = 0.0006; **** *p* < 0.0001.

**Figure 8 ijms-22-04549-f008:**
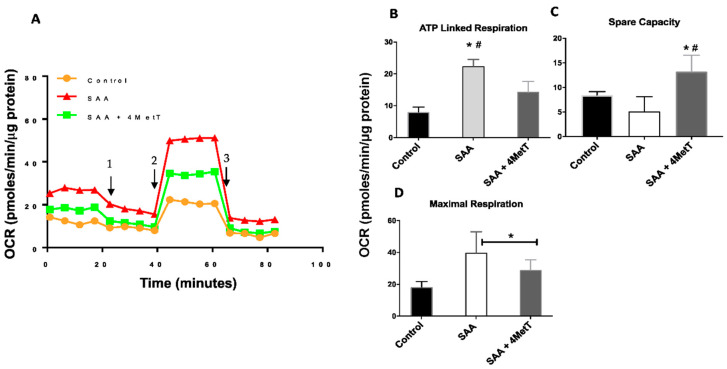
Effect of SAA and 4-MetT on cultured primary HAEC mitochondrial respiration. (**A**). Oxygen consumption rate profile (OCR); arrows indicate injection points for 1: oligomycin; 2: FCCP; 3: Rotenone/Antimycin. (**B**). ATP linked respiration (**C**). Spare respiratory capacity (**D**). Maximal respiration. Data represent mean ± SD, *n* = 5 per group. All data was normalised against corresponding total cellular protein * Different to control in the absence of SAA, *p* < 0.05. ^#^ Different to SAA-treatment group, *p* ≤ 0.001.

**Figure 9 ijms-22-04549-f009:**
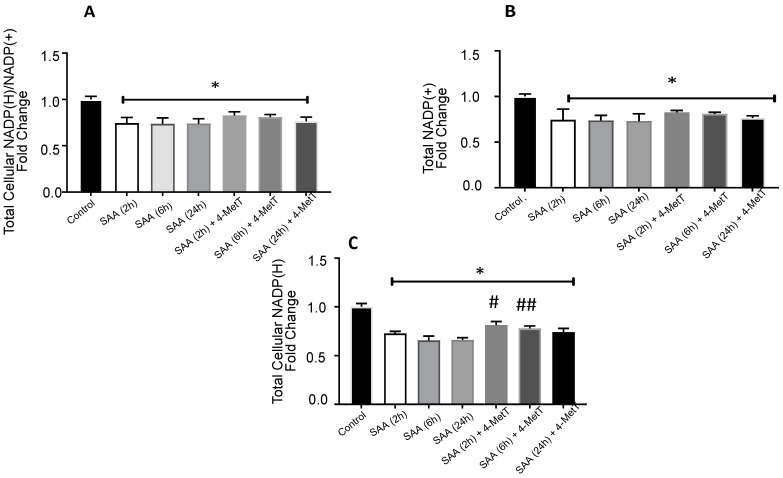
Effect on the NADP(H)/NADP(+) system in response to SAA and 4-MetT in HAEC. (**A**). Ratio of NADP(H): NADP(+) (**B**). Individual NADP(+) (**C**). Individual NADP(H). *n* = 3 SAA (2 h), SAA (2 h) + 4-MetT, *n* = 6 all other groups. * Different to control in the absence of SAA; * *p* < 0.0001. ^#^, ^##^ Different to SAA-treatment group, *p* < 0.005.

## Data Availability

Not applicable.

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
