# Peer review of "Efficacy of the Piperidine Nitroxide 4-MethoxyTEMPO in Ameliorating Serum Amyloid A-Mediated Vascular Inflammation"

_ijms, 2021, doi:10.3390/ijms22094549_

Round 1

Reviewer 1 Report

This manuscript by Martin, N. et al. assesses the efficacy of 4-MetT in preventing/decreasing the pro-inflammatory effects of SAA on primary human aortic endothelial cells. They demonstrate that SAA increases the expression of adhesion molecules, NF-kB and Caspase-1. In addition, SAA promotes the production of ROS, reduces the mitochondrial membrane potential and disturbs the mitochondrial respiratory potential. Conversely, pre-treating HAECs with 4-MetT prevents (sometimes partially) the effects mediated by SAA.   

While the manuscript provides interesting data, minor revisions and modifications are needed to warrant its acceptance in this journal. Furthermore, substantial addition to the paper (albeit relatively easy experiments) would improve its quality and impact immensely.

I have the following comments/questions:

  1. Figure 1 is barely visible both in print and in the PDF. The contrast of the figure should be adjusted. In addition, there is no clear difference between B and C. If the authors want to draw any conclusion, a quantifiable characterisation should be performed. Perhaps counting the number of nuclei per FOV should demonstrate the difference has cobblestone ECs are usually more dense.
  2. Figure 2: the cytoskeleton of a cell is composed of microfilaments, intermediate filaments and microtubules. Have the authors assessed the impact of SAA and 4-MetT on the other components of the cytoskeleton? Are the proteins linking the membrane to the cytoskeleton differentially regulated (FAK, talin, paxillin, vinculin, etc.)? The structural integrity of a cell is usually maintained by balancing in the contractile forces and the adherent tensions. As such, it would also be interested to look at the actomyosin. As it stands, the authors try to draw conclusion from superficial observations; simple additional experiments would allow a better understanding of the impact of SAA and 4-MetT.  
  3. As mentioned by the authors in the discussion, little is known about the role of P120 and other junctional molecules in atherosclerosis. As such, it would be interesting to assess the impact of SAA on both tight junction and adherens junction molecules (membrane molecules and intracellular adaptor molecules). This could shed light on the different morphology observed in figure 1 and could also link their observation in figure 2. Overall, this would improve this article.
  4. All experiments are conducted by pre-treating with 4-MetT. Has the effect of 4-MetT alone on the HAEC been assessed? If not, this should be done.
  5. The authors mention that 4-MetT could be a potential therapeutic approach for treating cardiovascular pathologies. However, none of the experiments performed here were carried out with a therapeutic approach in mind. As far as we know, treating the cells with 4-MetT after SAA could have no significant beneficial effect. These experiments should be performed or the text should be modified to reflect the data observed.
  6. The discussion is very long for a paper of this caliber. For example, the first three paragraph of the discussion simply state again and again the results without bringing any major additional information. Whereas other paragraphs are almost a review of the literature, mentioning information that are not necessary for comprehension of the significance of the results. Therefore, I would advise the authors to simplify the discussion. In addition, there are a few grammatical errors, missing determiners, poorly constructed sentences, etc. in the discussion.

Minor comments:

  1. Line 32: “a” should be added between is and key
  2. Line 52: a comma is needed after “in a recent study”
  3. Line 102-105: the sentence contains “support retention of…” twice. The sentence should be rephrase for clarity.
  4. Figure 3: the deltadelta on the axis of the figures are missing.
  5. Figure legend of figure 3 mention SAA for 6h and 24h, however, no data at 24h is shown.
  6. Line 149 refers to “this gene”. Since only VCAM is tested by WB and it is not the last gene mentioned in the text above, I would change the sentence to improve the clarity.
  7. Line 271-273: The sentence is confusing. “cells are expressing more stress”, “more energy to yielding”. It should be rephrased for clarity.
  8. Line 286-287 is confusing. What does “SAA response to acute phase response” mean? “and significant elevation” of SAA? The sentence should be rephrased for clarity.
  9. Line 288: replacing “to” by “of”
  10. Line 289: add a comma before “and increase number”
  11. Line 291: delete the superfluous “with simultaneous …of”, to improve clarity.
  12. In the discussion, the abbreviation “4-MetT” is inconsistently written.
  13. Line 315: a comma after study is missing.
  14. Line 323: SOD abbreviation not defined previously.

Final thoughts: I find the results shown in this manuscript to be interesting and deserving to be published. However, the quality and the impact of the article would be significantly improved if the authors can address my comments and performed the additional experiments suggested.  

Author Response

Reviewer 1

Thank you for providing very constructive comments and feedback. We have addressed the comments points wise and where required, have added new text/information in the revised version of the manuscript.

  1. Figure 1 is barely visible both in print and in the PDF. The contrast of the figure should be adjusted. In addition, there is no clear difference between B and C. If the authors want to draw any conclusion, a quantifiable characterisation should be performed. Perhaps counting the number of nuclei per FOV should demonstrate the difference has cobblestone ECs are usually more dense.

We have revised figure 1 with improved contrast. As suggested, We also quantified the number of cells per treatment group that showed non-significant difference between the SAA and 4-MetT group. However, cell morphology changed and tended to become elongated in the presence of SAA compared to cells treated with 4-MetT or vehicle as control.    The new text is added in the revised manuscript (Lines 87-90).

  1. Figure 2: the cytoskeleton of a cell is composed of microfilaments, intermediate filaments and microtubules. Have the authors assessed the impact of SAA and 4-MetT on the other components of the cytoskeleton? Are the proteins linking the membrane to the cytoskeleton differentially regulated (FAK, talin, paxillin, vinculin, etc.)? The structural integrity of a cell is usually maintained by balancing in the contractile forces and the adherent tensions. As such, it would also be interested to look at the actomyosin. As it stands, the authors try to draw conclusion from superficial observations; simple additional experiments would allow a better understanding of the impact of SAA and 4-MetT.  

With thanks, we completely agree with the reviewer. Undoubtedly the investigations of other markers (as suggested) would add significant meaning to the cell morphology results. However, the focus of this manuscript is the inhibition of the oxidative stress and inflammatory markers by 4-MetT.  While, the cell cytoskeleton markers were not investigated further the data presented here does offer some notion that goes to an explanation for the phenotypic change observed.  The latter certainly requires further study to prove the pathways for SAA-mediated cytoskeletal changes, but this will be considered in future studies as it is not the focus for this study.

  1. As mentioned by the authors in the discussion, little is known about the role of P120 and other junctional molecules in atherosclerosis. As such, it would be interesting to assess the impact of SAA on both tight junction and adherens junction molecules (membrane molecules and intracellular adaptor molecules). This could shed light on the different morphology observed in figure 1 and could also link their observation in figure 2. Overall, this would improve this article.

As pointed out by the reviewer, we also measured V cadherin and ESAM levels, but they were not significantly different among the groups thus data was not presented. However, they are now uploaded as supplementary material but do not add significantly to the explanation of morphological changes observed.  

  1. All experiments are conducted by pre-treating with 4-MetT. Has the effect of 4-MetT alone on the HAEC been assessed? If not, this should be done.

Unfortunately, 4MetT alone was not tested in these experiments and is one the limitations of this study. We have added this limitation in the discussion section (lines 390-394) now reading “Though, we have shown promising results of cyclic nitroxides in combating the SAA induced inflammation, the effect of 4-MetT alone on the cells and exact mechanism of action of this drug was not studied in these experiments which warrants further mech-anistic studies both in vitro as well as in vivo to explore whether the efficacy of 4-MetT can be translated in vivo.”.

We note however, that the available literature demonstrates that nitroxides are well tolerated by cells and when delivered in vivo as reviewed previously by our group (Antioxid Redox Signal 2020 Oct 1;33(10):689-712).

  1. The authors mention that 4-MetT could be a potential therapeutic approach for treating cardiovascular pathologies. However, none of the experiments performed here were carried out with a therapeutic approach in mind. As far as we know, treating the cells with 4-MetT after SAA could have no significant beneficial effect. These experiments should be performed, or the text should be modified to reflect the data observed.

As suggested, we have rephrased the text in the revised version (lines 25-27) that now reads:

“Subject to further validation in in vivo settings; these outcomes suggest its potential as a therapeutic in the setting of cardiovascular pathologies where elevated SAA and endothelial dysfunction is linked to enhanced CVD”

  1. The discussion is very long for a paper of this caliber. For example, the first three paragraph of the discussion simply state again and again the results without bringing any major additional information. Whereas other paragraphs are almost a review of the literature, mentioning information that are not necessary for comprehension of the significance of the results. Therefore, I would advise the authors to simplify the discussion. In addition, there are a few grammatical errors, missing determiners, poorly constructed sentences, etc. in the discussion.

As suggested, where required, we have reduced the text by 30% in the revised version.

Minor comments:

  1. Line 32: “a” should be added between is and key

We have corrected this error (line 32)

  1. Line 52: a comma is needed after “in a recent study”

Corrected (line 51)

  1. Line 102-105: the sentence contains “support retention of…” twice. The sentence should be rephrased for clarity.

We have revised the sentence (lines 105-108). New information reads “Though difficult to interpret, we speculate that 4-MetT supports retention of cell structure through normalising alpha tubulin expression and distribution, thus providing improved cytoskeletal integrity, cellular appearance and rate of cell proliferation in the presence of SAA-induced cellular stress”.

  1. Figure 3: the deltadelta on the axis of the figures are missing.

We have updated figure 3

  1. Figure legend of figure 3 mention SAA for 6h and 24h, however, no data at 24h is shown.

This error is corrected and as suggested by reviewer 2 the repetitive information of methodology is removed from the legends. 

  1. Line 149 refers to “this gene”. Since only VCAM is tested by WB and it is not the last gene mentioned in the text above, I would change the sentence to improve the clarity.

The sentence is revised (lines 151-152) now reading “Next, we investigated if expression of most relevant gene (VCAM-1) involved in atherosclerosis was translated into respective change in protein.

  1. Line 271-273: The sentence is confusing. “cells are expressing more stress”, “more energy to yielding”. It should be rephrased for clarity.

The sentence is revised (lines 264-266) and now reads “These data are in accord with the mitochondrial respiration experiments indicating that cells are expressing more stress and tend to consume more energy that leads to higher generation of ROS”.

  1. Line 286-287 is confusing. What does “SAA response to acute phase response” mean? “and significant elevation” of SAA? The sentence should be rephrased for clarity.

The sentence is revised (lines 274-275) and now reads “ Thus, significant increases in the circulating levels of SAA in response to APR is of direct clinical importance”

  1. Line 288: replacing “to” by “of”

The text containing this error was deleted from discussion.

  1. Line 289: add a comma before “and increase number”

The text containing this typo was deleted from discussion.

  1. Line 291: delete the superfluous “with simultaneous …of”, to improve clarity.

The text containing this information was deleted from discussion

  1. In the discussion, the abbreviation “4-MetT” is inconsistently written.

Thanks for pointing out. We have made consistent appearance (4-MetT) throughout the text.

  1. Line 315: a comma after study is missing.

Corrected (line 328)

  1. Line 323: SOD abbreviation not defined previously.

Full name “superoxide dismutase” is added now (line 299).

Reviewer 2 Report

In the manuscript entitled “Efficacy of the piperidine nitroxide 4-methoxyTEMPO in ameliorating serum amyloid A-mediated vascular inflammation” the authors demonstrate that 4MetT reverses pro-inflammatory phenotype in HAECs induced by SAA. The manuscript is scientifically sound and I have little comments on the scientific merit. The below comments are more geared towards improving the reading experience of the manuscript.

Abstract/Introduction:

Line 9: Intracellular redox balance – should this be inbalance?

Line 32: Include “a” before “key component

Line 45-47: sentence needs to be rewritten. E.g. “Accumulating evidence suggests that elevated serum SAA levels may be associated with promoting pathology, however not all research studies support these findings”. Then provide examples in order: all the studies supporting the role of SAA in promoting various pathologies first and those not supporting this notion second. Currently it is a bit mixed.

Line 48: “lesions” should be “lesion”

Line 59: “remain” should be “remains”

Line 59-61: Correct grammar in sentence: “SAA has been shown to be produced by endothelial/smooth muscle cells involved in atherogenesis and SAA was reported to modulate macrophage and monocyte function.”

Line 69: “better marker” than what? Otherwise state “… is a promising marker…” or “… may be a good marker…”

Materials and Methods:

Revisit heading titles and numbering under this section.

For every experiment, make sure to include the biological and technical replicates.

Line 458: “purchases” should be “purchased”

Line 466: “distill” should be “distilled”

Line 468-470: rewrite sentence so it is grammatically correct

Line 473: title not italic like other similar headings

Line 483: sentence needs to end with a bracket and point.

Line 484: title does not match what is described in this section. I suggest including first sentence in section 4.2.2 and moving the remainder of the paragraph to the “western blot” section. The very last sentence could be included under title 4.2.

Line 518: Does this need to be 3.5 microliters? It currently states 3.5 L. This is the case throughout this section. Likely symbol not coming through. Also, the degree symbol looks like a ‘0’ at several locations in the manuscript.

Line 520: 658nm/reaction? Can you give actual concentration?

Results

Overall the headings of the various sections should recapitulate the findings of the conclusions of the experiments; more along the lines of the titles of the figures rather than stating the technique used. The story should be a little more coherent and the transition between results should make sense – because of the results from experiment A, we did experiment B. The findings of experiment B led us to perform experiment C etc… So describing all the results from the gene expression analysis in one section might not be the most comprehensive way to present the data to the reader.  

Line 114: Gene expression study. Include a short rational why the authors decided to look at the expression of various molecules. Why NFkB? What is the rational for looking at VCAM-1/E-selectin/ICAM-1? Also, might want to reconsider the order in which the various gene expressions are discussed in this section. Might want to start with the expression of JAM-C and catenin as this logically follows Figure 1. What is the relationship between JAM-C/catenin and the pro-inflammatory proteins? There should be a logical transition from one to the other.

Line 120. The conclusion sentence should be more precise. The experiment showed that SAA induces NFkB expression and that 4-MetT can prevent the SAA-mediated increase in NFkB expression. The experiment was not set-up to proof that 4-MetT prevents the pro-inflammatory action of SAA.

Line 140: add letters A-F to Figure 3. Also, delta symbols are not visible within the figure.

Line 166: remove “notably”.

Line 169-170: Where do you show that p-NFkB is up?

Line 174: why are you looking at cGMP?

Line 180: techniques used are described in detail in the methods section. No need to repeat in detail in the figure legends.

Line 216-220: Can you explain the rotenone column?

Line 257: can you indicate in fig 8A at which points oligomycin, FCCP, antimycin A and rotenone were added to the reaction

Author Response

Reviewer 2

We thank the reviewer for such constructive comments that we have addressed point wise and where required added the text in the revised manuscript.  

Abstract/Introduction:

Line 9: Intracellular redox balance – should this be inbalance?

This is corrected to “imbalance” in the revised version of the manuscript (line 9)

Line 32: Include “a” before “key component

As suggested “a” is added (line 32)

Line 45-47: sentence needs to be rewritten. E.g. “Accumulating evidence suggests that elevated serum SAA levels may be associated with promoting pathology, however not all research studies support these findings”. Then provide examples in order: all the studies supporting the role of SAA in promoting various pathologies first and those not supporting this notion second. Currently it is a bit mixed.

As suggested, we have combined the studies which contradict the notion that SAA promotes the pathology and those which favour. We have rephrased the sentence now reading (lines 45-56). “Although, accumulating evidence suggests that elevated serum SAA levels are associated with promoting pathology, the data is inconclusive. For example, De Beer et. al. [16],…….

Line 48: “lesions” should be “lesion”

Corrected (line 48)

Line 59: “remain” should be “remains”

Corrected (line 59)

Line 59-61: Correct grammar in sentence: “SAA has been shown to be produced by endothelial/smooth muscle cells involved in atherogenesis and SAA was reported to modulate macrophage and monocyte function.”

We have rephrased the sentence now reading (lines 59-61) “SAA is produced by endothelial/smooth muscles cells involved in atherogenesis and SAA also modulates macrophages and monocyte’s function

Line 69: “better marker” than what? Otherwise state “… is a promising marker…” or “… may be a good marker…”

 Corrected to “promising” maker (line 69)

Materials and Methods:

Revisit heading titles and numbering under this section.

As suggested, all sections are revisited and made consistent

For every experiment, make sure to include the biological and technical replicates.

This information is added to methods sections (line 402)

Line 458: “purchases” should be “purchased”

Corrected (line 397)

Line 466: “distill” should be “distilled”

Corrected (line 405)

Line 468-470: rewrite sentence so it is grammatically correct

Revised (lines 406-407)

Line 473: title not italic like other similar headings

All titles have been revisited and made italic

Line 483: sentence needs to end with a bracket and point.

Corrected (line 422)

Line 484: title does not match what is described in this section. I suggest including first sentence in section 4.2.2 and moving the remainder of the paragraph to the “western blot” section. The very last sentence could be included under title 4.2.

As suggested, we have moved the text under western blot section (lines 472-478). 

Line 518: Does this need to be 3.5 microliters? It currently states 3.5 L. This is the case throughout this section. Likely symbol not coming through. Also, the degree symbol looks like a ‘0’ at several locations in the manuscript.

That seems a printing issue and have been corrected.  

Results

Overall the headings of the various sections should recapitulate the findings of the conclusions of the experiments; more along the lines of the titles of the figures rather than stating the technique used. The story should be a little more coherent and the transition between results should make sense – because of the results from experiment A, we did experiment B. The findings of experiment B led us to perform experiment C etc… So describing all the results from the gene expression analysis in one section might not be the most comprehensive way to present the data to the reader.  

As suggested, we have revised the results headings now recapitulating the findings of the experiments in relevant figures.  Continuously throughout the manuscript we have highlighted the text indicting transition from one experiment to another with justification.

Line 114: Gene expression study. Include a short rational why the authors decided to look at the expression of various molecules. Why NFkB? What is the rational for looking at VCAM-1/E-selectin/ICAM-1? Also, might want to reconsider the order in which the various gene expressions are discussed in this section. Might want to start with the expression of JAM-C and catenin as this logically follows Figure 1. What is the relationship between JAM-C/catenin and the pro-inflammatory proteins? There should be a logical transition from one to the other.

The reason for measuring NFkB is based on its strong transcriptional regulation of several pro-inflammatory mediators and adhesions molecules (i.e. VCAM-1, ICAM-1 and E-selectin) that play key role in atherosclerotic lesion development. The data of NFkB was presented first because this is the main key player that will lead to the transcription and over expression of all other discussed inflammatory markers. Once inflammatory response is established (evident through higher NFkB) this would modify the junctional complexes and the proteins involved in junction formation as is the case here. Thus, we believe that establishing the fact that inflammatory response has occurred was first thing to investigate as without this response expecting a change in junctional complexes will be a weak assumption. 

We have added new text in the gene analysis section (lines 115-120). Gene expression of NF-kB, a common transcriptional factor of inflammatory mediators and adhesions molecules (e,g., VCAM-1, ICAM-1 and E-selectin) were studied by qPCR (Figure 3). Once an inflammatory response is established to added SAA (as evidenced by increased expression of NF-κB), modification of the junctional complexes and the proteins involved in junction formation are anticipated to be impacted and for this reason we investigated the levels of both NF-κB and downstream adhesion molecules.

Line 120. The conclusion sentence should be more precise. The experiment showed that SAA induces NFkB expression and that 4-MetT can prevent the SAA-mediated increase in NFkB expression. The experiment was not set-up to proof that 4-MetT prevents the pro-inflammatory action of SAA.

We would like to clarify that the goal of these experiments was to establish the anti-inflammatory role of 4MetT against SAA induced pro-inflammatory changes that we have shown here. Thus, same was concluded given the strong inhibitory effects of 4MetT against SAA induced inflammatory markers.

Line 140: add letters A-F to Figure 3. Also, delta symbols are not visible within the figure.

Thanks for highlighting this error. We have updated figure 3.

Line 166: remove “notably”.

Corrected (line 167)

Line 169-170: Where do you show that p-NFkB is up?

Thanks again for highlighting this error. These results refer to the caspase experiments. We have updated and highlighted the text (lines 168-173).

Line 174: why are you looking at cGMP?

Cyclic guanosine monophosphate (cGMP) is an indirect marker of NO activity. Nitric oxide (NO) is a vasodilator and has significant importance in vascular pathology. Lower NO is a risk factor to vasoconstriction. Apoptotic activity of Caspase-1 is related to nitric oxide concentrations in the cell. For example, inhibition of NO has shown increased caspase-1 mediated inflammatory response i.e. conversion of pro IL-1β to active IL-1β as well as stimulation of IFN gamma inducing factor. Thus, cGMP levels were measured to indirectly relate the level of NO bioavailability.

Line 180: techniques used are described in detail in the methods section. No need to repeat in detail in the figure legends.

As suggested where required methodology details are removed from the figure legends. 

Line 216-220: Can you explain the rotenone column?

Rotenone was used as positive control which has known role to impair cellular respiration. This was run in parallel to compare with SAA group thus reconfirming SAA is inducing cellular changes in the treated cells and the experimental conditions are working.

The relevant text on Rotenone is added in the figure legend (lines 209-211).

Line 257: can you indicate in fig 8A at which points oligomycin, FCCP, antimycin A and rotenone were added to the reaction

As suggested, we have indicated the points when modulators were injected in the experiments as per manufactures instructions (Seahorse Biosciences). The injection points are indicated with arrows and relevant information is added in the figure legend (line 253).